# New Strategies for Biocontrol of Bacterial Toxins and Virulence: Focusing on Quorum-Sensing Interference and Biofilm Inhibition

**DOI:** 10.3390/toxins15090570

**Published:** 2023-09-15

**Authors:** Hua Zhang, Zhen Zhang, Jing Li, Guangyong Qin

**Affiliations:** 1Henan Key Laboratory of Ion Beam Bio-Engineering, College of Physics, Zhengzhou University, Zhengzhou 450000, China; zhanghua802233@163.com; 2School of Food and Biological Engineering, Henan University of Animal Husbandry and Economy, Zhengzhou 450046, China; lj5213318@163.com; 3School of Agricultural Sciences, Zhengzhou University, Zhengzhou 450000, China; qinguangyong@zzu.edu.cn

**Keywords:** biocontrol, bacterial toxins, virulence, probiotics, medicinal plants

## Abstract

The overuse of antibiotics and the emergence of multiple-antibiotic-resistant pathogens are becoming a serious threat to health security and the economy. Reducing antimicrobial resistance requires replacing antibiotic consumption with more biocontrol strategies to improve the immunity of animals and humans. Probiotics and medicinal plants have been used as alternative treatments or preventative therapies for a variety of diseases caused by bacterial infections. Therefore, we reviewed some of the anti-virulence and bacterial toxin-inhibiting strategies that are currently being developed; this review covers strategies focused on quenching pathogen quorum sensing (QS) systems, the disruption of biofilm formation and bacterial toxin neutralization. It highlights the probable mechanism of action for probiotics and medicinal plants. Although further research is needed before a definitive statement can be made on the efficacy of any of these interventions, the current literature offers new hope and a new tool in the arsenal in the fight against bacterial virulence factors and bacterial toxins.

## 1. Introduction

Bacterial toxins are described as being products of microbial metabolism; they are one of the creative strategies that bacteria have established for survival. Generally, endotoxins and exotoxins should be differentiated. Bacterial endotoxins are the main component of the cell wall or outer membrane of Gram-negative bacteria. Endotoxins cannot be released when the bacteria are alive; when the cells die and dissolve or when the bacteria are destroyed artificially, they are released; thus, they are called endotoxins. The chemical composition of endotoxins includes phosphoric acid, polysaccharides and proteins. The main ingredient is lipopolysaccharide (LPS) [1], a heat-stable amphiphilic molecule [2]. Lipid A is a toxic component of LPS and can mediate the binding of LPS to the host LPS-binding protein (LBP). In turn, the LPS-LBP complex binds to a cluster of differentiation 14 (CD14) and toll-like receptor 4 (TLR4) proteins on the surface of macrophages [3]. This binding can promote proinflammatory cytokine synthesis and release, which then inappropriately activates host complement cascades and coagulation pathways. This ultimately causes epithelial and endothelial cell damage, septic shock, hypotension and potentially fatal organ failure [4,5].

Exotoxins are mainly soluble toxic proteins that are secreted by microbes into the surrounding media [6]. Their toxicity is highly unstable in the presence of heat and certain chemicals and is easily undermined. Bacteria that produce exotoxins are mainly Gram-positive bacteria, such as *Staphylococcus aureus* (*S. aureus*), *Streptococcus* and *Bacillus tetanus* [7,8,9]. Some Gram-negative bacteria, such as *Pasteurella multocida* and virulent *Escherichia coli*, can also produce exotoxins [10,11]. Exotoxins can damage cell membrane structures, thus affecting the host cell surface. They can also block essential cellular processes to affect intracellular targets, e.g., intracellular trafficking, translation, signal transduction and actin cytoskeleton rearrangement [12]. Exotoxins include pore-forming toxins (PFT), membrane-breaking toxins and intracellular exotoxins that penetrate host cells in order to act.

Although the exotoxins and endotoxins of bacteria are completely different in terms of structure and mechanism of action, they can cause varying degrees of serious damage to the body. In order to infect a host, bacteria must develop a mechanism to combat the host’s immune system so that they can survive the harsh environment in the host, so different virulence factors are expressed at different times during the infection. Virulence factors are microbial structures and biomolecules that enable pathogens to achieve colonization, invasion and persistence within susceptible hosts [13]. Quorum-sensing (QS) systems indicate bacterial cell-to-cell communication processes. QS can activate many important processes in order to survive in the host and help establish pathogenesis. Therefore, the pathogenic mechanism of bacterial toxin infection in the host is one of the most valuable targets for the development of anti-virulence drugs [14,15]. QS and biofilms are both correlated processes that significantly affect pathogenic and physiological processes. Some anti-virulence strategies have been oriented to disturb biofilms by disrupting extracellular matrix production and bacterial adhesion and by disintegrating existing biofilms [16,17,18]. Additionally, toxin neutralization via probiotics facilitates an effective strategy to mitigate pathogens’ virulence. This is because pathogens use secreted toxins to colonize the host and evade host immune responses. 

Drug-resistant infections have become an increasingly severe problem worldwide. Thus, alternative methods (e.g., anti-virulence therapy that modulates the production of bacterial toxins or virulence factors) are needed to address antimicrobial-resistant strains. This comprehensive review reveals the biocontrol approaches being employed to quench the pathogen QS systems, attenuate bacterial toxins and disrupt biofilm formation (Figure 1).

## 2. Bacterial Toxin Neutralization

Shiga toxins (Stx1 and Stx2) are the main factors responsible for the virulence of *Escherichia coli* (STEC), inhibiting protein synthesis in cells with a nucleus and playing a role in the development of hemolytic uremic syndrome (HUS) and hemorrhagic colitis [19]. Even though STEC separated from patients with hemorrhagic colitis contains Stx1 and Stx2, human disease complications are more frequently attributed to Stx2. The study conducted by Carey et al. revealed that the most significant reduction in *stx2A* expression was observed after a 2-h co-incubation of *E. coli* O157:H7 with *Bacillus thermophilum*, *Lactobacillus rhamnosus*, and *Pediococcus pentosaceus*. Their findings indicate that the downregulation of *stx2A* is possibly associated with the pH effect of organic acids from probiotic bacteria [20]. Additionally, engineering probiotics and *E. coli* strains can achieve the expression of Gb3 receptor mimics on their surfaces, which absorb and neutralize Stx2 in vitro. Piglets that receive oral administration of bacteria expressing Gb3 analogs are protected from lethal challenge by virulent STEC strains [21]. (Figure 2A). The antimicrobial activity of *Saccharomyces cerevisiae* can be attributed to the elimination of secretory toxins, extracellular protease production, the stimulation of immunoglobulin A secretion, the hydrophobicity of cell surfaces, and autoaggregation ability [22]. In 1999, Czerucka et al. demonstrated that cholera toxin-induced secretion can be prevented by *Saccharomyces boulardii* (*S. boulardii*) in rat jejunum. They found a 120-kDa protein in a *Saccharomyces boulardii*-conditioned medium. This protein can inhibit cholera toxin-induced cAMP in intestinal cells. *S. boulardii* inhibited Cl^−^ secretion in vitro via both Ca^2+^- and cAMP-mediated signaling pathways [23] (Figure 2B). Brandão et al. showed that *S. boulardii* CNCM I-745 was able to adhere cholera toxin to its cell wall, i.e., this was another mechanism to reduce the cholera toxin [24].

It is well known that *Clostridium perfringens* type A (*C. perfringens* type A) is a typical foodborne pathogen. This bacterium can generate over 16 toxins. These toxins can induce intestinal and histotoxic infections in animals and humans [26]. The primary toxins include alpha toxin and enterotoxin. The findings of Kawarizadeh et al. showed that *B. coagulans* can inhibit the expression and growth of the alpha toxin gene in *C. perfringens* type A and thus reduce apoptosis and cytotoxicity against HT-29 cells. Therefore, *B. coagulans* can be applied in the prevention and treatment of *C. perfringens* type A gut infections [27]. As a foodborne natural compound, piceatannol has been commonly identified in various medicinal plants, vegetables, and fruits. Wang et al. demonstrated that piceatannol can prevent perfringolysin O (PFO) from damaging human intestinal epithelial cells, and LDH release decreased by 38.44% at 16 µg/mL in cytotoxicity tests. In this study, it was concluded that piceatannol could impede PFO’s pore-forming activity via direct binding, thereby changing PFO’s spatial conformation and inhibiting its oligomer formation. Ultimately, decreasing pore-forming activity occurred [28] (Figure 3A).

Streptolysin O (SLO) is an exotoxin from *Streptococcus pyogenes*. SLO can induce eukaryotic cell lysis, enabling *Streptococcus pyogenes* to evade phagocytosis and clearance via neutrophils and activate inflammatory bodies. Ruas-Madiedo et al. found that extracellular polymeric matrix (EPS) production in *lactobacilli* and *bifidobacteria* could antagonize bacterial pathogens’ toxicity. The EPS produced by *Bifidobacterium longum* NB667 against SLO’s hemolytic activity were assessed using rabbit erythrocytes. EPS NB667 caused the greatest decrease in hemolysis in erythrocytes [29]. Although the mechanisms of protection have not been completely revealed, we can hypothesize that bacterial EPS can act as toxin-scavenger agents by blocking receptors on the surfaces of eukaryotic cells. Guo et al. stated that luteolin can prevent SLO-induced cytotoxicity and changes to cell membrane permeability in HEp-2 cells. Luteolin inhibited SLO dissolution in erythrocytes by binding SLO with high affinity, suppressing oligomer formation, and affecting its conformational stability [30] (Figure 3B).

Botulinum neurotoxins (BoNTs) present a significant public health and safety risk due to their status as highly potent natural toxins. Both the BoNT serotype A complex and holo-toxin possess the capability to bind to and traverse intestinal epithelial cells, thus facilitating their systemic transportation through the bloodstream. According to Lam et al., probiotics such *S. boulardii*, *Lactobacillus acidophilus*, *L. s reuteri*, and *L. rhamnosus* LGG can prevent the binding and internalization of BoNT serotype A in mammalian cells. The related mechanism involved the competitive inhibition of host cell membrane receptors by probiotic strains and BoNT/A instead of the non-specific binding of the toxin to the probiotic or BoNT/A degradation [31]. 

*Aggregatibacter actinomycetemcomitans* (Aa) can induce inflammatory mediators in gingival epithelial cells. Then, Aa evades immune responses via internalization and the production of leukotoxin (*LtxA*) and cytolethal-distending toxin (*CdtB*) in non-phagocytic cells [32,33]. When Aa was grown with *Lactobacillus gasseri* and *Lactobacillus salivarius* cell-free supernatants (CFSs), two genes were downregulated in a time-dependent manner, according to Nissen et al.’s analysis of the *LtxA* and *CdtB* levels [34]. Another study has shown that expression of Aa profile can be changed by Lactobacilli postbiotics in a strain-specific manner. *L. rhamnosus* LR32 and *Lactobacillus acidophilus* LA5 CFS downregulated the transcription of *CdtB* and *LtxA* [35].

## 3. Overview of Quorum-Sensing and Biofilm Formation

### 3.1. Quorum Sensing

Bacterial QS modulates gene expression based on bacterial population density via cell-to-cell communication [36]. For Gram-positive bacteria, oligopeptides are the main signaling molecules in the QS system and act as autoinducers (AIs). *N*-acyl-homoserine lactones (AHLs) are the primary signaling molecules serving as AIs in Gram-negative bacteria [37,38]. AHLs consist of the amino acid derivative homoserine lactone (HSL). In addition, the furanosyl borate diester molecule AI-2 is another signaling molecule and is present in Gram-positive and Gram-negative bacteria [39]. Microbial toxins are vital components of the virulence factors of microbes [40], and their production mechanisms are controlled by the QS signaling system [41,42]. Disrupting bacterial QS activity or this communication system can reduce microbial virulence [43]. The degradation or inactivation of QS signaling molecules is recognized as quorum quenching (QQ) or QS inhibition. There are several ways to stop a QS signal, including: (1) ceasing the synthesis of signaling molecules; (2) interfering with the binding of signaling receptors in bacterial cells; (3) enzymatically destroying or inactivating signaling molecules to stop accumulation from reaching a certain threshold; and (4) blocking target genes that should be activated by QS signals [44,45]. By employing anti-toxin therapeutic strategies, it is possible to interfere with intercellular communication and monitor the presence of infectious bacteria without impeding their proliferation. Consequently, this approach effectively hinders the development of antibiotic resistance [46,47]. Therefore, the recent development of non-toxic, broad-spectrum QQ drugs from microorganisms and plants has been greatly advantageous.

### 3.2. Biofilm Formation

Biofilm formation is a crucial virulence factor. It is also a common phenomenon among microorganisms. Following biofilm formation, microbes can diffuse and colonize other environments [48]. These bacteria include pathogenic bacteria. They can act as a reservoir for persistent infections and can contribute to the emergence of widespread chronic illnesses and antibiotic resistance. Thus, it is difficult to treat those illnesses. Biofilms exhibit complex three-dimensional structures. The bacteria become embedded in their EPS networks, which consist of polysaccharides, extracellular DNA (eDNA), proteins, lipids [49], and other organic compounds from the surrounding environments or secreted by bacteria [50]. Extracellular polymeric substances (EPS) serve as a barrier and boundary between external environments and microbial communities, dominating bacterial adhesion. In addition, released eDNA from the lysis of bacterial cell subpopulations participates in the biofilms’ attachment, aggregation, and stabilization [51]. Adhesion to highly hydrophobic surfaces can even be promoted by eDNA [52]. Biofilms with high water content allow for nutrient transport within biofilms for bacterial survival [53]. Furthermore, three signaling systems may affect biofilm formation: (1) QS, (2)3′,5′-cyclic diguanylic acid (c-di-GMP) [54] and (3) two-component signaling systems (TCS) [55]. QS is strongly correlated with biofilm formation, and the genes and surfactant molecules involved in biofilm formation are regulated by QS [56]. Many marketed antibiotics cannot affect biofilms, especially when the biofilms are produced by resistant bacteria. Thus, various attempts have been made to obtain compounds from plants, fungi, animals, bacteria and viruses [57]. Understanding the mechanisms of bacterial biofilm formation can facilitate the identification of potential targets for compounds that only impact biofilms without killing bacteria. For instance, endotoxins from bacterial cells that disperse from biofilms can be neutralized by anti-virulence agents, thus preventing or minimizing the detrimental effects of the inflammatory response of the host to bacterial infections.

## 4. Anti-Virulence Treatment Strategies

### 4.1. Secondary Metabolites of Bacteria as Quorum-Quenching Agents

Directly damaging targeted cells by producing antimicrobial metabolites and controlling adaptive mechanisms, such as biofilm formation, can induce antagonisms between bacteria and fungi or among bacteria. The QQ phenomenon allows for such an antagonistic approach. The co-development of secondary metabolites is vital in bacterial evolution. These secondary metabolites are capable of disrupting QS signaling molecules and attenuating other microorganisms’ virulence. The ability to disrupt QQ or QS signaling molecules may have evolved in QS bacteria in order to eliminate or repurpose their own QS signaling molecules or those of the microorganisms co-inhabiting competitive environments [58]. Molecular evolution may have occurred in bacteria for AHL degradation in order to use AHL as a single carbon and nitrogen source or as a shield against the bacteria that produce antibiotics [59].

Lactic acid bacteria (LAB) have a significant effect on human life. In different countries, many types of LAB are utilized to produce traditional fermented food. Lactobacilli (types of LAB) are essential components of the normal intestinal microbiota of animals and humans. Primarily, with regard to the benefits of LAB, they can generate lactic acid and other substances (e.g., antimicrobial peptides and hydrogen peroxide). These substances can inhibit the growth of other bacteria [60,61]. LAB can also have immunomodulatory effects as symbionts within gastrointestinal tracts [62]. The potential activity of lactobacilli as QQ agents has also been identified. LAB can possibly impact QS, mediated by furanosyl borate diester (also recognized as AI-2) and acylated homoserine lactones (AHL, HSL).

Valdéz et al. have shown that *Lactobacillus plantarum* ATCC 10241 has the ability to inhibit the production of AHLs, which are QS signaling molecules capable of inducing the virulence factors of *Pseudomonas aeruginosa* (*P. aeruginosa*) PA100 [63]. In a study by Chandla et al., the CFS of Lactobacillus fermentum PUM (an indigenous potential probiotic) was able to inhibit protease, elastase, pyocyanin, pyochelin, and the virulence factors and motility of *P. aeruginosa* PAO1 alone or in combination with zingerone [64]. In a study conducted by Rana et al., the acid fraction of CFS (*L. lactis*, *L. fermentum* and *L. rhamnosus*) displayed a noteworthy decrease (*p* ≤ 0.05) in the auto-inducer AHL levels of *P. aeruginosa* PAO1 coupled with a decline in elastase activity [65] (Figure 4A). The capacity of different Lactobacillus strains to impede the function of different AHLs has been demonstrated. For example, Kampouris et al. focused on addressing overgrowth on filter membranes and found that LAB can hinder N-Hexanoyl-L-homoserine lactones (6-HSLs). These LAB were isolated from activated sludge, specifically *L. plantarum*, and subsequently enclosed within alginate beads. *L. plantarum* strain SBR04MA produced the most significant outcomes. Within 9 h, all of the 6-HSLs were degraded by this strain [66]. An *L. acidophilus* 30CS cell extract was found to significantly inhibit the AI-2 activity of *E. coli* O157:H7, without effects on EHEC growth [67]. According to Han et al., the anti-QS abilities of *L. plantarum* Z057 against *V. parahaemolyticus* were found to be remarkable. It was observed that Z057-E could effectively impede EPS production as well as inhibit the QS signaling molecule AI-2 and extracellular protein secretions [68].

Many studies have been dedicated to interactions of lactobacilli with QS signals from Gram-positive bacteria besides than their ability to inhibit QS signaling in Gram-negative bacteria via interactions with various AHL types. The studies have focused on the inhibition of QS in *S. aureus.* Yan et al. demonstrated that biosurfactants from *P. acidilactici* and *L. plantarum* reduced the AI-2 expression in *S. aureus* in a dose-dependent manner [69]. In addition, Liu et al. found that lipopeptide (a biosurfactant produced by *Bacillus subtilis*) influenced the QS system in *S. aureus* by regulating the AI-2 activity [70] (Figure 4B). Regarding other Gram-positive pathogenic bacteria, the QS (luxS) system and AI-2 in *C. difficile* were suppressed upon addition of the heat-treated supernatant *L. fermentum* Lim2. This was attributed to the suppression of *agr* genes [71]. 

Probiotics can inhibit the activity of pathogenic bacteria and their adhesion to surfaces using different mechanisms. They can inhibit the survival of biofilm pathogens and biofilm formation, disrupt biofilm integrity/quality and ultimately cause biofilm eradication. The competitive adhesion of probiotics to human tissues and medical equipment can inhibit colonization by harmful bacteria. In addition, probiotics can inhibit pathogenic biofilm formation by reducing biofilm biomass and environmental pH.

An important human pathogen, methicillin-resistant *S. aureus* (*MRSA*) can induce serious infectious diseases. *L. acidophilus* inhibits lipase production and biofilm formation in *S. aureus* as reported by Sikorska et al. Their effects were mediated by the direct exclusion of cellular competition and by the production of bacteriocin-like inhibitors and short-chain fatty acids [72]. Melo et al. demonstrated that *L. fermentum* TCUESC01, derived from cocoa seeds, effectively hindered biofilm formation of *S. aureus.* The mechanism of hindrance occurred via the suppression of two genes, *icaA* and *icaR*, which are known to significantly impact biofilm synthesis after the discharge of soluble molecules [73]. Squarzanti et al. conducted an evaluation of the activity of *L. johnsonii* LJO02 (DSM 33828) and *L. rhamnosus* LR06 (DSM 21981) CFSs against MDR *S. aureus* (ATCC 43300) in two different media: an innovative animal derivative-free broth (TIL) and the conventional animal derivative-based MRS medium. All CFSs decreased the viability and metabolic activity of *S. aureus*. TIL was more effective in reducing *S. aureus* biofilm formation and stimulating LAB metabolism compared to MRS [74]. Mastitis is one of the most critical and multi-factorial diseases influencing dairy cows. In the dairy industry, it is crucial to treat and manage mastitis for public health reasons, economic growth, and animal welfare. Sevin and colleagues demonstrated that the postbiotics, which include various fatty acids, vitamins, and organic acids, were derived from the milk microbiota. These postbiotics were found to be a complex mixture of metabolic by-products secreted by *L. sakei* EIR/BG-1. These postbiotics could be employed as promising agents to prevent mastitis due to their antibacterial and antibiofilm activity against *Streptococcus agalactiae* ATCC 27956, *Streptococcus dysgalactiae* subsp. *dysgalactiae* ATCC 27957, and *MRSA* ATCC 43300 [75] (Figure 5).

As a chronic infectious disease with multiple factors, dental caries are initiated by bacterial biofilm formation, primarily by *Streptococcus mutans*. *L. rhamnosus*-derived biosurfactants markedly reduce the biofilm-forming capacity of *S. mutans* by suppressing genes associated with biofilm formation, such as *gtfB*/*C* and *ftf*. *L. rhamnosus*-derived biosurfactants have strong anti-adhesive activity and are thus appropriate candidates as new-generation microbial antiadhesive agents [76]. Emine et al. reported that postbiotic mediators (PMs) of *Lactiplantibacillus plantarum* EIR/IF-1 separated from infant feces exhibited the most significant inhibitory (pH-dependent) effect against *S. mutans* (ATCC 25175). Decreased cell viability and significant variations in biofilm formation were also proved using confocal laser scanning microscopy and scanning electron microscopy on glass coverslips and in vitro on human tooth surfaces. Furthermore, the expression of *comA*, *comX*, and *gtfC* was downregulated by sub-MIC values of the PMs without any significant inhibitory effects on growth [77]. It has been demonstrated that Lactobacillus sp. has the ability to manage dental caries and deter tooth decay. This potential anti-caries effect could be attributed to several factors: (1) the reduction of cell adherence and pre-formed biofilms; (2) the inhibition of *Streptococcus mutans* growth, which is mainly due to peroxide production and organic acid generation; (3) the immunomodulatory effects resulting from the inhibition of IL-10 production and the induction of IFN-γ production; and (4) the downregulation of several *Streptococcus mutans* virulence genes, such as EPS-producing genes (*gtf BCD* and *sacB*), acid tolerance genes (*atpD* and *aguD* genes), and quorum-sensing genes (*vicKR* and *comCD*) [78]. Dawwam et al. investigated the antibiofilm and antibacterial effects of *L. plantarum* and *L. acidophilus* on multidrug-resistant *E. coli* from urine samples. In addition, *Lb. plantarum* and *L. acidophilus* strains reduced the ability of uropathogenic *E. coli* (UPEC) to develop biofilms by 39.63% and 56.3%, respectively. The variations in the gene expressions of *csgA*, *crl*, and *csgD* were significantly downregulated after separate treatment with *L. plantarum* and *L. acidophilus* suspensions [79]. Hossain et al. examined the effect of four sub-MICs of *L. plantarum* M.2 and *L. curvatus* B.67 (1/2, 1/4, 1/8, and 1/16 MIC) on the formation of *L. monocytogenes* biofilm on various food contact surfaces. The higher sub-MICs (1/2, 1/4, and 1/8 MIC) of both postbiotics considerably impeded biofilm formation at 30 °C for 24 h (*p* < 0.05) on silicone rubber, plastic, rubber gloves, and an MBEC™ biofilm device. However, the lower sub-MIC (1/16 MIC) did not have any significant inhibitory effects (*p* > 0.05). RT-qPCR results showed that the expression of several target genes associated with *L. monocytogenes* biofilm formation was downregulated by postbiotics. These target genes included those genes involved in QS (*agrA*), motility (*flaA*, *fbp*), and virulence (*hlyA*, *prfA*), which significantly affect biofilm formation [80]. The lactic acid bacteria categories and their molecular mechanisms are presented in Table 1.

### 4.2. Secondary Metabolites of Medicinal Plants as Quorum-Quenching Agents

Over the past decade, an incredible amount of research has been conducted to identify plants with anti-QS properties and biofilm inhibition activity. Phytochemicals can regulate bacterial AHL synthesis and in turn suppress QS. Various natural extracts are thought to inhibit QS by disrupting AHL activity via competition with AHLs due to their structural similarity and/or by accelerating the degradation of the LuxR/LasR receptors of AHL molecules [81].

The chemical classes from the investigated anti-virulence medicinal plants mainly include phenolic derivatives, terpenoids, and alkaloids.

Phenolic compounds belong to a large group of secondary plant metabolites. Phenol (the simplest phenolic compound) is an aromatic ring with a single hydroxyl group. Polyphenols are composed of two or more such phenolic units and exhibit a broad structural diversity. Flavonoids belong to the polyphenol family and are water-soluble polyphenolic molecules with 15 carbon atoms. It has been reported that baicalein, a type of flavone, exhibits the potential to enhance proteolysis of the *Agrobacterium tumefaciens* QS signaling receptor TraR in *Escherichia coli* cells when present at millimolar concentrations. Additionally, this particular flavone demonstrates inhibitory effects on biofilm formation dependent on QS in *P. aeruginosa* PAO1 at micromolar concentrations [82,83]. It has been shown that microorganisms utilize biofilms for attachment and development on surfaces. Once biofilm formation is completed, the bacteria inside are more difficult to target. Hydrophilic flavonoids can be used as antibiofilm agents via various mechanisms [84]. This is consistent with the findings of Vikram et al., they found that kaempferol, rutin, apigenin, sinensetin, and quercetin can reduce the biofilms of *V. harveyi* and *E. coli* [85]. 

As an essential constituent of rich tea, epigallocatechin exhibits antibiofilm properties against *Salmonella typhimurium* via the modulation of *luxS* and *diA* gene expression levels [86]. Epigallocatechin exhibits inhibitory effects on quorum sensing and biofilm development in *Listeria monocytogenes* [87], *Burkholderia cepacia* [88], and *Eikenella corrodens* [89].

Quercetin, which is widely distributed in fruits and vegetables, has diverse and multiple effects against QS depending on the tested bacterial strains. Quercetin also shows antagonistic effects on bacterial signaling. In addition, the suppression of biofilm formation by quercetin was found in *Escherichia coli* O157:H7 [85]. Quercetin has inhibitory effects on QS-controlled virulence factors, e.g., pyocyanin and violacein in *P. aeruginosa* PAO1, *P. aeruginosa* PAF79, and *Aeromonas hydrophila* WAF38 [90]. According to Ouyang et al., the *P. aeruginosa* strain can be effectively inhibited by quercetin, thus demonstrating antibiofilm properties. Additionally, quercetin at a concentration of 16 μg/mL has the ability to decrease the expression levels of *rhll*, *rhlR*, *lasl,* and *lasR* [91]. Quercetin can also interact with transcriptional regulation of *LasR* in *P. aeruginosa* to inhibit QS circuitry [92].

Other flavonoids have also been found to exhibit significant anti-QS properties. For instance, a study conducted by Chemmugil et al. revealed that morin effectively impeded the formation of biofilms in *S. aureus* by reducing its motility, spreading ability, and production of EPS [93]. Up to 80% of biofilm formation, the expression of adhesion-related genes, and the activity of *S. aureus* sortaseA (SrtA) can be inhibited by kaempferol at a concentration of 64 μg/mL [94]. Conversely, taxifolin can considerably diminish the production of elastase and pyocyanin in *P. aeruginosa* while not affecting bacterial growth. Moreover, this substance can lower the expression of specific QS-controlled genes (namely *rhlI*, *rhlR*, *phzA1*, *rhlA*, *lasI*, *lasR*, *lasA*, and *lasB*) in *P. aeruginosa* PAO1 [95]. As an isoflavone compound, genistein can be derived from most leguminous plant foods [96]. Aerolysin is an important virulence factor of pathogenic *Aeromonas.* Dong et al. found that genistein can inhibit biofilm formation and aerolysin generation, which is dose-dependent. When co-cultured with genistein, the transcription of QS-related genes (*ahyI* and *ahyR*) and the aerolysin encoding gene (*aerA*) were drastically downregulated [97]. In addition, luteolin and esculetin can also inhibit the production of biofilm and aerolysin to affect the pathogenesis of *Aeromonas hydrophila* [98,99].

As natural antioxidants in plants, phenolic acids (PAs) have an aromatic ring with methoxy or OH groups. The number of methoxy or OH groups determine their diversity. Their antibacterial activities are associated with their chemical structures [100,101]. Joshi et al. showed that salicylic acid (SA) and cinnamic acid affected the QS mechanism of *P. carotovorum ssp. brasiliense* and *P. aroidearum* and thus changed the expression of their bacterial virulence factors. Although the expression of QS-related genes increases over time in control treatments, the exposure of bacteria to non-lethal concentrations of cinnamic acid or SA can suppress the expression of QS genes (i.e., *PC 1_1442* (*lux R* transcriptional regulator) and *luxS* (a component of the AI-2 system)), *expI*, and *exp R* [102]. Chlorogenic acid (CA) is also well known as 3-caffeoylquinic acid and coffee tannic acid and can reduce the LPS content of *P. aeruginosa* P1 to encourage the outer membrane’s detachment. The expression of the major genes (*LPxB* and *LPxC*) in LPS biosynthesis can also be downregulated by CA [103]. 

Liu and colleagues conducted a study on the effects of vanillic acid (VA) against *Vibrio alginolyticus*, a pathogen that affects fish. VA was found to significantly reduce the ability of *V. alginolyticus* to form biofilms, its mobility, and its production of exotoxins such as protease and exopolysaccharide. In addition, VA downregulated the expression of genes (*asp*, *luxR sypG*, *lafA*, *lafK*, *fliS,* and *fliK*) associated with biofilm formation and virulence in *V. alginolyticus* when used at sub-inhibitory concentrations [104]. Kannan et al. discovered that the growth of biofilms and the production of lipase, hemolysin, and elastase through QS were hindered by rosmarinic acid (RA) at a concentration of 750 µg/mL in *A. hydrophila* strains. Additionally, RA has the ability to decrease the expression of virulence genes such as *lip*, *ahyB*, *ahh1*, and *aerA* [105].

Tannins can affect toxin production and are significant for bacterial pathogenicity and virulence. Jailani et al. reported that polyphenol tannic acid can efficiently inhibit the biofilm formation and planktonic growth of *A. tumefaciens* on abiotic (polystyrene and nitrocellulose) and plant root surfaces and can inhibit the virulence traits (e.g., protease activity, EPS secretion, swimming motility, and cell surface hydrophobicity). One mechanism for antibacterial and subsequent biofilm inhibition may be attributed to iron chelation via tannic acid because of galloyl groups. The reduced expressions of *chvE* and *dnaK* may be the other mechanism [106]. Tannic acid has been found to prevent biofilm formation in *E. coli* [107] and *S. aureus* [108]. Its anti-biofilm effects in *S. aureus* are mainly related to the production of protein immunodominant staphylococcal antigen A (IsaA). IsaA is a putative lytic transglycosylase that is capable of cleaving the β-1,4-glycosidic bonds between *N*-acetylglucosamine (GlcNAc) and *N*-acetylmuramic acid (MurNAc) in the peptidoglycan layer. Cleaving the peptidoglycan layer can reduce the biofilm thickness of *S. aureus* [109] (Figure 6A). It has been shown that pomegranate extracts can effectively inhibit the growth and toxin production of the multidrug-resistant Clostridium difficile hypervirulent strain (NAP1/027/BI). The inhibitory effects of pomegranate extracts on clostridial toxin B (TcdB) could be due to ellagitannins (specifically punicalagin) [108]. Li et al. reported that punicalagin can dependently suppress the expression of QS-related genes (*sdiA* and *srgE*) in *Salmonella* and the production of violacein by *Chromobacterium violaceum* [110]. Even though *Salmonella* is not capable of synthesizing AHLs on its own, it can perceive AHL signals from other microorganisms via SdiA (a LuxR homolog) in *Salmonella*. *Salmonella* can also control a broad range of traits [111] (Figure 6B). Sivasankar et al. discovered that tannic acid exhibited noteworthy anti-QS activity, effectively impeding the swarming of *Salmonella Paratyphi* A and *Salmonella Typhi* regulated via QS at a concentration of 400 μg/mL. Tannic acid demonstrates the ability to inhibit QS and associated virulence factors without causing harm or exerting continuous pressure on the viability of the pathogen [112]. *Salmonella* sp. has the SdiA and luxS/AI-2 QS systems [113].

Terpenoids are derived from the condensation of isopentane or isoprene units and mainly include sesquiterpenoids, monoterpenoids, and diterpenoids. Linalool is a monoterpenoid with a tertiary alcohol group, which is derived from many plants, including J. Presl (Lauraceae), *Lavandula* spp. (Lamiaceae), *Cannabis sativa* L. (Cannabaceae), and *Cinnamomum camphora* (L.) [114,115,116]. Linalool derived from Coriandrum sativum exhibits antibacterial properties against *Acinetobacter baumannii* (*A. baumannii*). Its mechanism of action against *A. baumannii* involves disrupting quorum sensing, impeding biofilm development, and inhibiting bacterial adhesion. Linalool effectively hinders the formation of biofilms by dispersing them and preventing their growth [117]. Some monoterpenes can be subject to 100% reduction in biofilm biomass. For example, the bacterial viability of *S. aureus* treated with geraniol decreases at 1 mg/mL, and its biofilm formation shows a decrease at concentrations between 0.5 and 4 mg/mL [118]. The antibacterial potential of some terpenoids, such as that of myrtenol, has also been studied. This bicyclic monoterpene alcohol shows good efficacy against *MRSA* via inhibitory effects on biofilms and anti-virulence activity against the main virulence factors (staphyloxanthin, α-hemolysin, slime, autolysin, and lipase) [119]. With regard to another monoterpene component (namely bicyclic sesquiterpene, known as β-caryophyllene), it prevents biofilm formation and cell growth when tested against *Streptococcus mutans* and reduces the expression of *gtf* genes [120]. 

Monoterpenes and sesquiterpenes, as well as their oxygen derivatives, are the most conventional types of citrus essential oils (EOs) [121]. The widely acclaimed anti-biofilm activity of bioactive compounds in citrus Eos enables Eos to be useful for new applications of pest control drugs, antimicrobial agents, and herbicides [122,123]. The bactericidal activity evaluation of (+)-limonene, a primary component found in citrus essential oils, demonstrates its ability to effectively combat biofilm formation and dental caries caused by *S. pyogenes* and *S. mutans*, respectively. This monoterpenoid exhibits a dose-dependent response, inhibiting the adhesion properties of *S. pyogenes* biofilms while also downregulating *vicR* gene expression in *S. mutans*, ultimately leading to reduced acid production [124]. The EOs of juniper, clary sage, marjoram, and lemon and their components are considered to be promising candidates in the prevention of biofilm formation and the AHL-mediated QS mechanism [125]. EOs from *E. radiate* and *E. globules* also show anti-QS properties. 1,8-cineole is the major component of *E. globulus* oil and is known as eucalyptol (63.81%). 1,8-cineole in *E. globulus* oil and limonene (68.51%) in *E. radiata* oil suppress the QS phenomenon by inhibiting QS-regulated violacein pigment production in bacteria without any effects on their growth [126]. *Dorema aucheri* Bioss. and *Ferula asafoetida* L. EOs at 25 μg/mL exhibit anti-QS activity against *P. aeruginosa.* Ferula completely inhibited the violacein production of *C. violaceum*, while Dorema only reduced the violacein production of *C. violaceum* to some extent at the same concentration. Pyoverdine, elastase, pyocyanin, and biofilm production were reduced under treatments of Ferula oil. Dorema oil diminished elastase and pyoverdine production while not affecting biofilm and pyocyanin production [127].

Alkaloid berberine is another compound exhibiting antibiofilm activity. It binds to EPS-associated amyloid proteins to affect the biofilm formation of *S. epidermidis*. It was found that berberine has bacteriostatic effects on *S. epidermidis*. Berberine can inhibit the formation of *S. epidermidis* biofilms at sub-minimal inhibitory concentrations. Berberine suppresses bacterial metabolism at concentrations of 15–30 μg/mL and can exhibit antibacterial effects and significantly inhibit the biofilm formation of *S. epidermidis* ATCC 35984 and its clinical isolate strain SE243S at moderate concentrations of 30–45 μg/mL [128]. The medicinal plant categories and their molecular mechanisms are presented in Table 2.

## 5. Conclusions and Discussion

The threat of antimicrobial resistance has motivated the scientific community to find effective solutions. Therefore, the biocontrol of pathogen virulence has been seen as a promising alternative approach which aims to inhibit the production or activity of vir-ulence factors and not affect general bacterial growth. The current studies of probiotics or natural plants are mainly aimed at the prevention of bacterial adhesion or biofilm formation, the interruption or inhibition of the work of bacterial secretion systems, and the downregulation of virulence. Other strategies include reducing or blocking quorum sensing, and via global or specific regulators, gene expression regulation was performed. Here, we review the possible mechanisms and research progress of probiotics (mainly lactic acid bacteria, yeast, bacillus. and bifidobacterium) in neutralizing bacterial toxins such as Shiga toxin, cholera toxin, alpha toxin, streptolysin O, and botulinum neurotoxin. These probiotics exhibit potential antibacterial activity while also disrupting the expression of genes associated with the quorum sensing system of pathogenic bacteria and influencing biofilm formation. The chemical classes from the investigated anti-virulence medicinal plants mainly included phenolic derivatives, terpenoids, and alkaloids. Relevant secondary metabolites from medicinal plants have biofilm-inhibitory activity and anti-quorum sensing (QS) characteristics. As an emerging therapeutic strategy, the development of biocontrol therapy still faces several challenges. The main limitations of studies on biocontrol of bacterial toxins and virulence include the paucity of human or animal studies; their ideal dose; the duration of supplementation; and the durability of their beneficial effects as well as their safety profile in the treatment of pathogenic bacteria.

It is clear that medicinal plant resources and probiotics are abundant. However, many aspects have not been studied. It should also be recognized that the working modes of many effective inhibitors remain unclear. Therefore, the study of their bioactive mechanisms can be taken as a future research direction.

To promote further progress, future studies need to solve one or more of the following aspects of the biocontrol of bacterial toxins: The determination of whether probiotics and medicinal plants can concurrently re-duce both pathogens and toxins produced by the pathogens.The development of probiotic-based oral vaccines to protect animals against contamination by toxins.The determination of whether the anti-toxin effects of probiotics and medicinal plants and extracts in vitro can be duplicated in vivo, especially in humans.

## Figures and Tables

**Figure 1 toxins-15-00570-f001:**
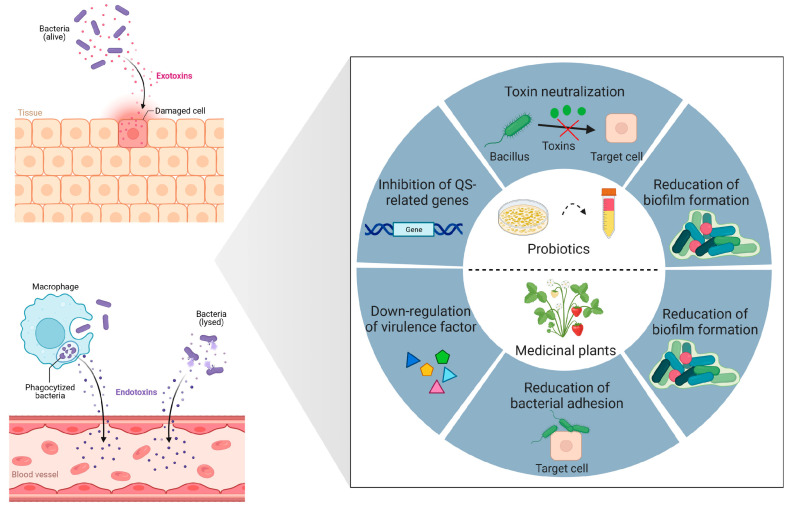
Biocontrol strategies for bacterial toxins and virulence. Probiotics and their secondary metabolites employ sophisticated anti-virulence strategies, including neutralization of bacterial toxins, repression of gene expression associated with quorum sensing (QS), and reduction of biofilm formation by pathogenic bacteria. Medicinal plants and their secondary metabolites have the ability to downregulate the expression of genes associated with virulence factors, reducing pathogen adherence to host cells, as well as inhibiting biofilm formation. Created with BioRender.com (accessed on 15 February 2023).

**Figure 2 toxins-15-00570-f002:**
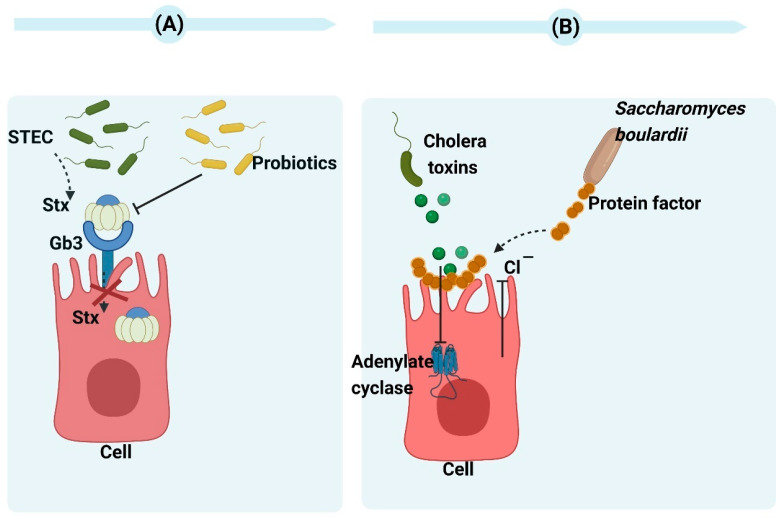
(**A**) The mechanism of probiotics action against Shiga toxins. By enabling the probiotic to express receptors for Shiga toxins, it can function as a biological mechanism that effectively eliminates toxins and minimizes tissue damage caused by infection [25]. (**B**) The mechanism of *Saccharomyces boulardii* CNCM I-745 action against cholera toxins. The protein produced by *Saccharomyces boulardii* CNCM I-745 has a molecular weight of 120 kDa and effectively inhibits adenylate cyclase activity as well as chloride secretion induced by cholera toxin. Moreover, it demonstrates the ability to bind with cholera toxins. Created with BioRender.com (accessed on 15 February 2023).

**Figure 3 toxins-15-00570-f003:**
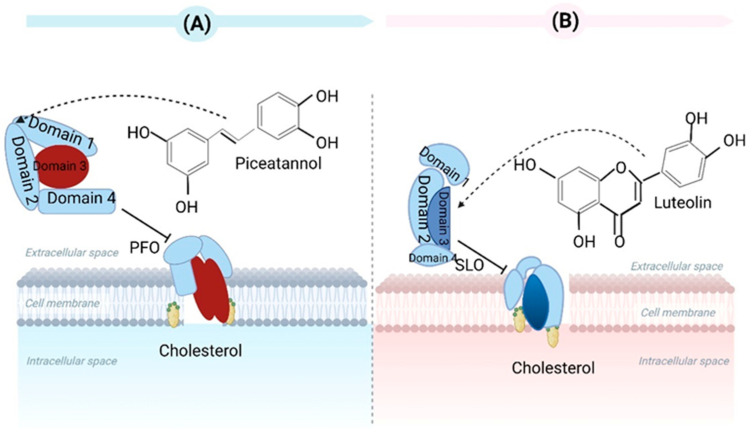
(**A**) Piceatannol has the ability to hinder the pore-forming function of PFO by directly interacting with *Clostridium perfringens.* (**B**) Luteolin has the ability to strongly bind to SLO that is generated by *Streptococcus pyogenes*, leading to alterations in its conformational stability and hindering the formation of oligomers. Created with BioRender.com (accessed on 15 February 2023).

**Figure 4 toxins-15-00570-f004:**
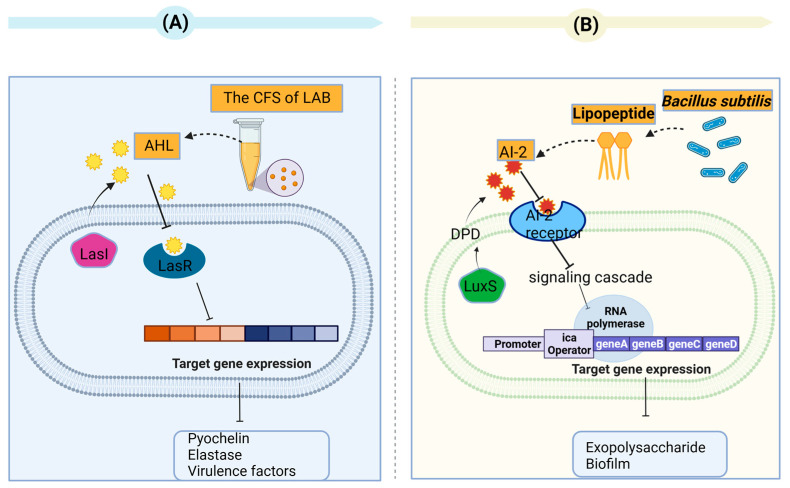
(**A**) The supernatant of lactic acid bacteria inhibits quorum sensing in *Pseudomonas aeruginosa.* The QS systems involve the signal synthases LasI and the receptors LasR in *P. aeruginosa.* The Las systems employ acyl-homoserine lactones (AHL), which are autoinducer signaling molecules. The Las system controls pyochelin, elastase, and virulence factors. (**B**) The lipopeptide produced by *Bacillus subtilis* affected the quorum-sensing (QS) system in *S. aureus* via regulation of the auto inducer 2 (AI-2). The QS systems involve the signal synthases LuxS and the AI-2 receptors in *S. aureus*. AI-2 derived from DPD (4,5-dihydroxy-2,3-pentanedione) synthesized by LuxS synthase in the S-adenosyl-methionine (SAM) recycling pathway.

**Figure 5 toxins-15-00570-f005:**
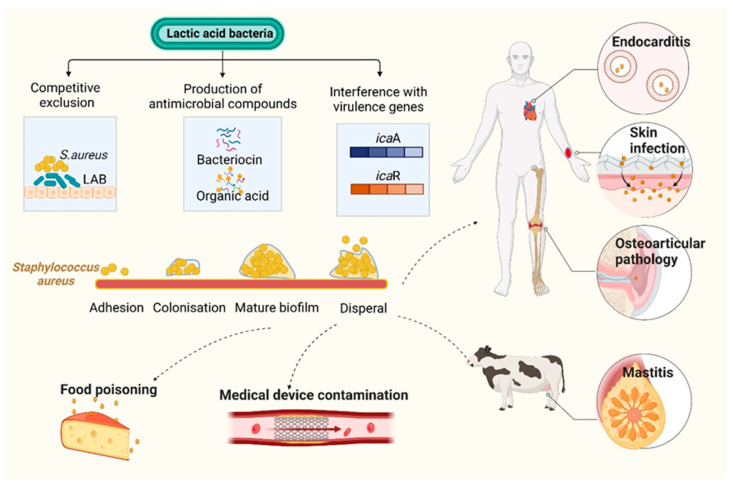
Mechanism of inhibition of *Staphylococcus aureus* biofilm by lactic acid bacteria (LAB) and their metabolites. The formation of biofilm proceeds through four different stages in *S. aureus*, which are: (1) adhesion of planktonic cells to the surface (either a biotic host or any abiotic surface); (2) colonization; (3) biofilm maturation; and (4) biofilm dispersal. *S. aureus* is a leading source of opportunistic infections, including those relating to skin, osteoarticular pathology, endocarditis, and contaminated introduced devices. In addition, it can cause food poisoning. LAB can inhibit formation of *S. aureus* biofilm via numerous mechanisms. For example: (1) Most LAB can inhibit *S. aureus* via the production of antimicrobial substances such as bacteriocins or organic acid; (2) LAB can compete with *S. aureus* for adhesion sites on the surfaces of planktonic cells, thereby preventing harmful colonization; (3) LAB can alter the expression of virulence genes associated with *S. aureus* biofilms. Figure created in BioRender (http://biorender.io, accessed on 15 February 2023).

**Figure 6 toxins-15-00570-f006:**
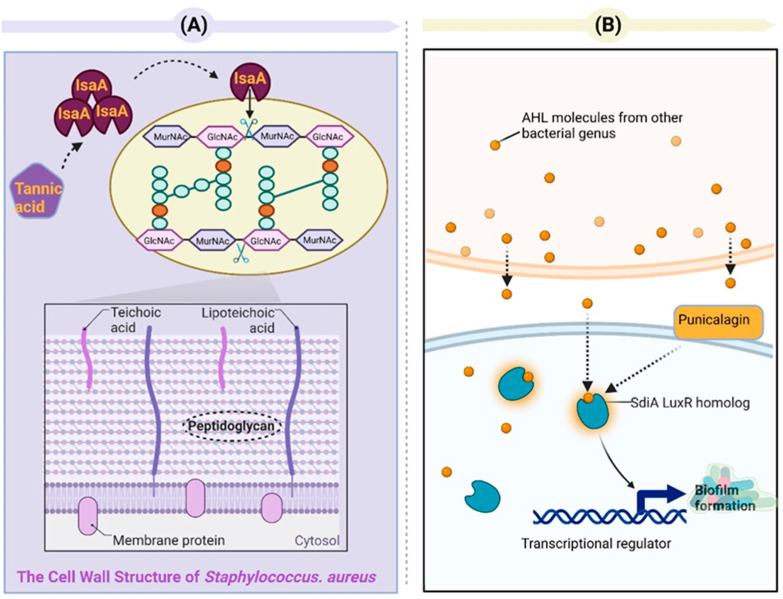
(**A**) Tannic acid exhibits inhibitory effects on the formation of *Staphylococcus aureus* biofilms via an IsaA-dependent mechanism. IsaA, a putative lytic transglycosylase, has been demonstrated to possess peptidoglycan cleavage activity via zymography analysis. Lytic transglycosylases belong to a distinct class of lysozyme-like enzymes that facilitate the breakdown of the *β*-1,4-glycosidic bond between *N*-acetylmuramic acid (MurNAc) and *N*-acetylglucosamine (GlcNAc). The expression of IsaA effectively hinders *S. aureus* from establishing biofilm structures. (**B**) Punicalagin inhibits the activation of QS-related genes (*sdiA*) in *Salmonella*. The luxS/autoinducer 2 (AI-2) system QS and SdiA QS system are present in *Salmonella* sp. [114]. SdiA, a LuxR-type receptor unique to *Salmonella*, responds to AHL signals produced by other species and regulates genes involved in various aspects of host colonization. Punicalagin downregulates the expression of the *sdiA* gene in *Salmonella*.

**Table 1 toxins-15-00570-t001:** Strains of lactic acid bacteria—quorum-quenching antagonists.

Lactic Acid Bacteria	Mechanism of Action in QS Systems	Bacteria	Study	Reference
*Lactobacillus plantarum* ATCC 10241	Inhibits acyl-homoserine lactone activity and decreased elastase and biofilm formation	*Pseudomonas aeruginosa* PA100	in vitro and burned-mouse model	Valdez 2005 [63]
*Lactobacillus rhamnosus* MTCC 5897, *Lactobacillus fermentum* MTCC 5898, *Lactococcus lactis* NCDC-309	Reduces levels of the auto-inducer AHL and causes a decline in elastase activity and a decrease in mRNA expression of *lasI* and *rhlI*	*Pseudomonas aeruginosa* PAO1	in vitro	Rana 2020 [65]
*Lactobacillus acidophilus* strain 30SC	Inhibits autoinducer-2 (AI-2) activity and represses biofilm formation	*Escherichia coli* (*EHEC*) O157:H7	in vitro	Kim 2018 [67]
*Lactiplantibacillus plantarum* Z057	Disrupts the protective biofilm layer, suppresses the communication signal molecule AI-2 involved in quorum sensing, and reduces the activity of genes associated with quorum sensing (*luxS*, *aphA*, and *opaR*), as well as hemolysin-related genes (*ToxS* and *ToxR*)	*Vibrio parahaemolyticus* ATCC 17802	in vitro	Han 2022 [68]
*Pediococcus acidilactici* 27167 *Lactobacillus plantarum* 27172	Reduces expression levels of biofilm-related genes (*cidA*, *icaA*, *dltB*, *agrA*, *sortaseA* and *sarA*) and interferes with the release of signaling molecules (AI-2) in QS systems	*Staphylococcus aureus* CMCC 26003	in vitro	Yan 2019 [69]
*Lactobacillus fermentum* Lim2	Reduces autoinducer-2 (AI-2) activity and suppresses quorum-sensing (*luxS*) and virulence factors (*tcdA*, *tcdB*, and *tcdE*)	*Clostridioides difficile* 027	in vitro	Yong 2019 [71]
*Lactobacillus fermentum* TCUESC01	Reduces formation of biofilm, increases *icaR* gene, and reduces *icaA* gene expression	*Staphylococcus aureus* CCMB262	in vitro	Melo 2016 [73]
*Lactobacillus rhamnosus* ATCC7469	Reduces the expression level of *gtfB*, *gtfC*, and *ftf* genes	*Streptococcus mutans* ATCC 35668 and 22	in vitro	Tahmourespour 2019 [76]
*Lactiplantibacillus plantarum* EIR/IF-1 *Lactiplantibacillus curvatus* EIR/DG-1 *Lactiplantibacillus curvatus* EIR/BG-2	Inhibits biofilm formation and downregulates expression of *gtfC*, *comA*, and *comX*	*Streptococcus mutans* (ATCC 25175)	ex vivo human tooth surfaces	OmerOglou 2022 [77]
*Lactobacillus reuteri* (ATCC 23272) *Lactobacillus plantarum subspecies plantarum* (ATCC 14917) *Lactobacillus salivarius* (ATCC 11741)	Reduces adherence of preformed biofilm and gene expression of glucan (*gtfB*, *gtfC*, *gtfD*) and fructan (*sacB*)	*Streptococcus mutans* (ATCC 25175)	in vitro	Wasfi 2018 [78]
*Lactobacillus acidophilus* (ATCC 4356) *Lactobacillus plantarum* (ATCC 14917)	Inhibits ability to form biofilms and downregulates expression of biofilm genes (*csgA*, *crl* and *csgD*)	Uropathogenic *E. coli*	in vitro	Dawwam 2022 [79]
*Lactobacillus curvatus* B.67 and *Lactobacillus plantarum* M.2	Suppresses levels of QS (*agrA*) and expression of motility-related genes (*flaA*, *fbp*) and virulence-associated genes (*hlyA*, *prfA*)	*Listeria monocytogenes* ATCC 19113	rubber gloves, plastic, silicon rubber surfaces and MBEC™ biofilm device	Hossain 2021 [80]

**Table 2 toxins-15-00570-t002:** Anti-quorum sensing effects of medicinal plants.

Class of Compound	Phytochemical	Mechanism of Action in QS Systems	Bacteria	Reference
Flavonoid	Baicalein	Reduces biofilm formation at 20 μM concentration	*Pseudomonas aeruginosa* PAO1	Zeng 2008 [83]
	Naringenin	Suppresses virulence genes (*vopD*, *vscO* and *vcrD*) and disrupts cell-cell signalling and biofilm formation	*Escherichia coli* O157:H7 ATCC 43895 and *Vibrio harveyi* BB120	Vikram 2010 [85]
	Licochalcone A and epigallocatechin-3-gallate	Downregulates the expression of QS-associated genes (*sdiA* and *luxS*) at sub-MIC concentration	*Salmonella Typhimurium*	Hosseinzadeh 2020 [86]
	Epigallocatechin	Reduces biofilm formation	*Listeria monocytogenes* (LMG 21263)	Nyila 2012 [87]
	Epigallocatechin	Reduces biofilm formation at 40 µg/mL concentration	*Burkholderia cepacia*	Huber 2003 [88]
	Catechin	Affects autoinducer 2-mediated quorum sensing and inhibits biofilm formation	*Eikenella corrodens* 1073	Matsunaga 2010 [89]
	Quercetin	Inhibits QS-controlled virulence factors such as violacein, elastase, and pyocyanin and biofilm formation at sub-MIC concentration	*Pseudomonas aeruginosa* PAO1 *Pseudomonas aeruginosa* PAF79 *Aeromonas hydrophila* WAF38	Al-Yousef 2017 [90]
	Quercetin	Inhibits biofilm formation and suppresses the production of virulence factors, such as pyocyanin, protease, and elastase, at a concentration of 16 μg/mL. Additionally, there was a decrease in the expression levels of genes associated with quorum sensing (*lasI*, *lasR*, *rhlI*, and *rhlR*)	*Pseudomonas aeruginosa* PAO1	Ouyang 2016 [91]
	Morin	Reduces biofilm formation, motility, and spreading and EPS production	*Staphylococcus aureus*	Chemmugil 2019 [93]
	Kaempferol	Prevents the formation of biofilms by inhibiting the activity of sortase A and downregulates the expression of adhesion-related genes (*clfA*, *clfB*, *fnbA,* and *fnbB*)	*Staphylococcus aureus* ATCC 29213™	Ming 2017 [94]
	Taxifolin	Diminishes levels of QS-regulated genes, including *lasI*, *lasR*, *rhlI*, *rhlR*, *lasA*, *lasB*, *phzA1*, and *rhlA*.	*Pseudomonas aeruginosa* PAO1	Vandeputte 2011 [95]
	Genistein	Decreases the production of aerolysin and biofilm formation at a dose-dependent manner and downregulates QS-related genes (*ahyI* and *ahyR*) and aerolysin encoding gene (*aerA*)	*Aeromonas hydrophila* XS-91-4-1	Dong 2021 [97]
	Esculetin	Inhibits the production of protease and hemolysin, and the formation of biofilms, downregulates QS-related and biofilm formation-related genes (*ahyI*, *ahyR*, *luxS*, *csgAB*, and *fleQ*), and negatively upregulates biofilm formation-related gene (*litR*)	*Aeromonas hydrophila* SHAe 115	Sun 2021 [98]
Phenolic acids	Cinnamic acid and salicylic acid	Inhibits the expression of QS genes, including *expI*, *expR*, *luxR*, and *luxS*; reduces the level of the AHL signal	*Pectobacterium aroidearum* PC1 *Pectobacterium carotovorum* ssp. *brasiliense* Pcb1692	Joshi 2016 [102]
	Chlorogenic acid	Downregulates the expression of major genes (*LPxB* and *LPxC*) in LPS biosynthesis	*Pseudomonas aeruginosa* P1	Su 2019 [103]
	Vanillic acid	Diminishes capacity to form biofilms, movement ability, and production of exotoxins (protease and exopolysaccharide), while suppressing the expression of genes associated with biofilm formation and virulence (*sypG*, *fliS*, *fliK*, *lafA*, *lafK*, *asp*, and *luxR*) when exposed to subinhibitory concentrations	*Vibrio alginolyticus*	Liu 2021 [104]
	Rosmarinic acid	Reduces QS-mediated hemolysin, lipase, and elastase production at 750 µg/mL concentration; downregulates virulence genes (*ahh1*, *aerA*, *lip*, and *ahyB*)	*Aeromonas hydrophila*	Devi 2016 [105]
Tannic	Tannic acid	Reduces biofilm formation; downregulates the adhesion-associated *exoR* gene, limited the iron supply	*Agrobacterium tumefaciens* GV2260	Jailani 2022 [106]
	Ellagic acid Tannic acid	Reduces biofilm formation	*Escherichiacoli* VR50 and F18	Hancock 2010 [107]
	Tannic acid	Reduces biofilm formation	*Staphylococcus aureus*	Payne 2013 [108]
	Punicalagin	Represses the expression of QS-related genes (*sdiA* and *srgE*)	*Salmonella Typhimurium* SL1344	Li 2014 [110]
Terpenoid	Linalool	Inhibits the formation of biofilms and disrupts pre-existing biofilms, alters adhesion properties, and interferes with the quorum sensing mechanism	*Acinetobacter baumannii*	Alves 2016 [117]
	Geraniol	Reduces biofilm biomass	*Staphylococcus aureus* ATCC 6538	Pontes 2019 [118]
	Myrtenol	Reverses the formation of mature biofilm; inhibits the production of key virulence factors such as slime, lipase, alpha-hemolysin, staphyloxanthin, and autolysin. Reduces the activity of the global regulator *sarA* and its influence on virulence genes (*agrA*, *icaA*, *icaD*, *fnbA*, *fnbB*, *clfA*, *cna*, *hla*, *hld*, *geh*, *altA,* and *crtM*).	Methicillin-resistant *Staphylococcus aureus*	Selvaraj 2019 [119]
	β-caryophyllene	Inhibits biofilm formation and reduces the expression of *gtf* genes	*Streptococcus mutans* KCTC 3065 (ATCC 25175)	Yoo 2018 [120]
	Limonene	Inhibits adhesion and prevents biofilm formation cascade	*Streptococcus pyogenes* (SF370 and 5 clinical isolates) *Streptococcus mutans* (UA159) *Streptococcus mitis* (ATCC 6249)	Subramenium 2015 [124]
	Ferula Dorema	Decreases pyocyanin, pyoverdine, elastase, and biofilm production Reduces pyoverdine and elastase production	*Pseudomonas aeruginosa* PAO1 (ATTC 15692)	Sepahi 2015 [127]
Alkaloid	Berberine	Inhibits biofilm formation at sub-MIC concentrations	*Staphylococcus epidermidis* 243 (SE243) *Staphylococcus epidermidis* ATCC 35984 and ATCC 12228	Wang 2009 [128]

## Data Availability

Not applicable.

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
