# Peer review of "New Strategies for Biocontrol of Bacterial Toxins and Virulence: Focusing on Quorum-Sensing Interference and Biofilm Inhibition"

_toxins, 2023, doi:10.3390/toxins15090570_

Round 1

Reviewer 1 Report (Previous Reviewer 2)

Dear Authors:

I evaluate your sincere effort for revision.  I hope this will be accepted soon.

Thank you for your great work.

Author Response

Thank you very much for your advice

Reviewer 2 Report (New Reviewer)

Major comments

·         The basic details about probiotics and their biocontrol activities were missing.

·         The mechanism of probiotics activities should be explained in detail as a separate subheading.

·         Similarly, basic details about plant bioactives are missing.

·         Bio controlling using probiotics and plant bioactives in various applications should be included in the manuscript, for example, its role in food industries, medicine, environment, etc.

·         Details of products/strategies for bio-controlling using probiotics and plant bioactives in day -to day life has to be included in the manuscript.

·         The practical difficulties in the focused approaches have to be included in the manuscript.

·         Include the limitations of the use of probiotics for bio-controlling.

Some of the minor issues:

The current figure does not convey anything in detail. Revise Figure 1 and add an explanation.

Explain the abbreviations when used the first time, for example. Line 77, what is STEC?

Similarly, use full genus name for bacterial strains when used for the first time in the manuscript. Line 81, 82, etc.

Author Response

Response to Reviewer 2 Comment

Ponit 1: The basic details about probiotics and their biocontrol activities were missing.  Similarly, basic details about plant bioactives are missing.

Response 1: We are not entirely sure what your question is referring to.

Ponit 2: The mechanism of probiotics activities should be explained in detail as a separate subheading.

Response 2: Probiotics possess the potential for exerting antimicrobial activity, and a depiction of their antibacterial mechanism is presented in the manuscript as figures.

Ponit 3: Bio controlling using probiotics and plant bioactives in various applications should be

included in the manuscript, for example, its role in food industries, medicine, environment, etc. Details of products/strategies for bio-controlling using probiotics and plant bioactives in day -to day life has to be included in the manuscript.

Response 3: In this review, we present some of the anti-virulence strategies that focus on quenching pathogen quorum sensing (QS) systems, disrupting biofilm formation, and neutralizing bacterial toxins. The review emphasizes the probable mechanism of action for probiotics and medicinal plants in countering bacterial infections. However, it is essential to note that further research is needed before definitive statements can be made regarding the efficacy of these interventions. Nonetheless, the current literature offers new hope and a valuable tool in the fight against bacterial virulence factors and toxins.

 Ponit 4:  The practical difficulties in the focused approaches have to be included in the manuscript.

         Include the limitations of the use of probiotics for bio-controlling.

Response 4: Thank you for your suggestion. We have revised. The main limitations of studies on biocontrol of bacterial toxins and virulence include the paucity of human or animal studies; their ideal dose; the duration of supplementation; and the durability of their beneficial effects as well as their safety profile in the treatment of Pathogenic bacteria.

Ponit 5: The current figure does not convey anything in detail. Revise Figure 1 and add an explanation.

Response 5: Thank you for your suggestion. We have revised.

Ponit 6: Explain the abbreviations when used the first time, for example. Line 77, what is STEC?

Response 6: Thank you for your suggestion. We have revised.

Ponit 7: Similarly, use full genus name for bacterial strains when used for the first time in the manuscript. Line 81, 82, etc.

Response 7: Thank you for your suggestion. We have revised.

Reviewer 3 Report (New Reviewer)

This article reviewed the use of natural compounds as agents to inhibit quorum sensing in bacteria as a mechanism to reduce pathogenesis. There are still grammatical errors in the paper that should be reviewed. I have attached a sheet that has specific recommendations line by line. The big ask I have is to talk about the limitation of all the studies reviewed in this article, that are all in vitro. Thus, it is unknown whether these compounds will be efficacious in vivo and also whether or not the effective dose can be  toxic to the hosts.

Author Response

Response to Reviewer 3 Comment

Abstract:

Ponit 1: Line 10: bacterial toxin-inhibiting strategies.

Response 1: Thank you for your suggestion. We have revised.

Ponit 2: Line 11: This review ----- strategies focused to quench.

Response 2: Thank you for your suggestion. We have revised.

Introduction:

Ponit 3: Line 26: Bacteria or toxins? Cannot be released.

Response 3: Bacteria endo-toxins.

Ponit 4: Line 32: of LPS to the host LPS-binding protein.

Response 4: Thank you for your suggestion. We have revised.

Ponit 5: Line 58: To extend host survival or survival in the host?

Response 5: Thank you for your suggestion. We have revised.

Ponit 6: Line 59: The sentence Therefore, pathogenesis serves ------- seems very out of place and out of context in this paragraph.

Response 6: Thank you for your suggestion. We have revised.

Ponit 7: Line 75: Figure 1 legend has no explanation just a title.

Response 7: We have revised. Figure 1. Biocontrol strategies for bacterial toxins and virulence. Probiotics and their secondary metabolites employ sophisticated anti-virulence strategies, including neutralization of bacterial toxins, repression of gene expression associated with quorum sensing (QS), and reduction of biofilm formation by pathogenic bacteria. Medicinal plants and their secondary metabolites have the ability to down-regulate the expression of genes associated with virulence factors, reducing pathogen adherence to host cells, as well as inhibiting biofilm formation.

Ponit 8: Line 77 and 78: STEC; HUS – the first time you put an abbreviation; you should spell it out.

Response 8: Thank you for your suggestion. We have revised.

Ponit 9: Line 82: Include genus names the first time you include it. For example, Bacillus thermophilum, Lactobacillus rhamnosus, and P. pentosaceus? Italicize bacterial and fungal names.

Response 9: Thank you for your suggestion. We have revised.

Ponit 10: Line 83: GG?

Response 10: GG has been deleted.

Ponit 11: Line 85: The expression of Gb3 receptor mimics on their surface can be achieved – I am not sure what that means.

Response 11: Many experimental approaches to the treatment of EHEC infection and/or HUS have been explored. One potential therapeutic approach for HUS is to neutralize the activity of Stx and several compounds have been developed for that purpose, including oral synthetic receptor mimics and specific anti-Stx antibody. In this fashion, a probiotic that expresses Stx receptor mimics on its surface was developed for treatment of active EHEC infection. The advantage of this particular probiotic is its ability to bind and absorb Stx within the intestinal lumen, therefore potentially increasing its utility as a post-exposure therapeutic agent. The probiotic is a recombinant E. coli R1 strain (CWG308) that contains a plasmid (pJCP-Gb3) encoding two Neisseria sp. galactosyltransferase genes that when expressed, create a cell surface mimic of the Stx receptor. The binding capacity of this recombinant strain for either Stx1 or Stx2 is approximately 10,000× greater than that of SYNSORB Pk (a synthetic carbohydrate receptor mimic previously developed for treatment of STEC infection), and this efficacy has been demonstrated by several in vivo protection studies in mice. This particular construct was effective at neutralizing most of the Stx2 variants; however, it was less efficacious against Stx2e, which binds preferentially to Gb4.

Ponit 12: Line 186: Spell out EPS.

Response 12: We have revised.

Ponit 13: Line 197: it should be are not is before regulated by QS.

Response 13: We have revised.

Ponit 14: Line 232: Lactobacillus has an incorrect space.

Response 14: We have revised.

Ponit 15: Line 376: Quercetin – no mention of where it is derived from.

Response 15: We have revised.

Ponit 16: Line 491 Either say to prevent or say in the prevention of

Response 16: We have revised.

Ponit 17: Discussion: The limitation of these studies (all done in vitro) has to be presented here. It is not known at all whether these compounds can be used in vivo in terms of their effectiveness as

well as toxicity to the patients.

Response 17: Thank you for your suggestion. We have revised.

Reviewer 4 Report (New Reviewer)

Dear authors,

I have carefully reviewed your article titled "New Strategies for Biocontrol of Bacterial Toxins and Virulence: Focusing on Quorum-Sensing Interference and Biofilm Inhibition," and I must commend you on the comprehensive analysis of the emerging biocontrol strategies to combat antimicrobial resistance.

I have noted a few technical errors and suggestions for improvement in your abstract. Please find the revised abstract below:

1. Abstract:

The overuse of antibiotics and the emergence of multiple-antibiotic resistant pathogens have posed a serious threat to health security and the economy. In order to mitigate antimicrobial resistance, it is imperative to shift from excessive antibiotic consumption to more effective biocontrol strategies that enhance the immunity of both animals and humans. Among these strategies, probiotics and medicinal plants have emerged as promising alternative treatments and preventative therapies for various bacterial infections.

In this review, we present some of the anti-virulence strategies that focus on quenching pathogen quorum sensing (QS) systems, disrupting biofilm formation, and neutralizing bacterial toxins. The review emphasizes the probable mechanism of action for probiotics and medicinal plants in countering bacterial infections. However, it is essential to note that further research is needed before definitive statements can be made regarding the efficacy of these interventions. Nonetheless, the current literature offers new hope and a valuable tool in the fight against bacterial virulence factors and toxins.

2. English: The review article requires extensive editing of English language.

3. Discussion and Conclusion: The discussion section can be elaborated, allowing for a critical analysis of the existing strategies and the formulation of futuristic plans. Furthermore, it is essential to include a conclusion section to summarize the key findings and implications of the study.

Overall, your article provides valuable insights into the potential of biocontrol strategies and their role in combating bacterial infections. I have no doubt that this contribution will serve as a vital reference for researchers and practitioners in the field of antimicrobial resistance.

Once you have made the necessary revisions, kindly resubmit the article, and I will be more than happy to reevaluate it. 

Thank you for your dedication to advancing scientific knowledge.

Extensive editing of English language required.

Author Response

Response to Reviewer 4 Comment

Ponit 1: I have noted a few technical errors and suggestions for improvement in your abstract. Please find the revised abstract below.

Response 1: Thank you for your suggestion. We have revised.

Ponit 2:  English: The review article requires extensive editing of English language.

Response 2: The manuscript has been thoroughly revised and rewritten by MDPI English Editing.

Ponit 3: Discussion and Conclusion: The discussion section can be elaborated, allowing for a critical analysis of the existing strategies and the formulation of futuristic plans. Furthermore, it is essential to include a conclusion section to summarize the key findings and implications of the study.

Response 3: Thank you for your suggestion. We have revised.

Round 2

Reviewer 3 Report (New Reviewer)

I am happy with the improvements you have made in the manuscript. A few minor adjustments that can be made are: 

Line 11: This review covers ----- should be a new sentence. 

Line 26: Shouldn’t it say “Endotoxins cannot be released when the bacteria are alive”

Line 88: Thank you for your explanation, but the sentence “the expression of Gb3 receptor mimics on their surface can be achieved” is still confusing the way it is written. May be rewrite it in a different way.

Please make sure everywhere you put names of bacteria/fungi, they are italicized.

There are still some grammatical errors that editing staff can hopfully correct.

Author Response

Ponit 1: Line 11: This review covers ----- should be a new sentence. 

Response 1: Yes, this is a new sentence.

Ponit 2: Line 26: Shouldn’t it say “Endotoxins cannot be released when the bacteria are alive”

Response 2: Thank you for your suggestion. We have revised.

Ponit 3: Line 88: Thank you for your explanation, but the sentence “the expression of Gb3 receptor mimics on their surface can be achieved” is still confusing the way it is written. May be rewrite it in a different way.

Response 3: Additionally, engineering probiotics and E. coli strains can achieve the expression of Gb3 receptor mimics on their surfaces, which absorb and neutralize Stx2 in vitro. Piglets that receive oral administration of bacteria expressing Gb3 analogs are protected from lethal challenge by virulent STEC strains.

Point 4: Please make sure everywhere you put names of bacteria/fungi, they are italicized.

Response 4: Thank you for your suggestion. we have also carefully checked the italicized writing of the names of bacteria and fungi.

Reviewer 4 Report (New Reviewer)

The authors have made substantial improvements to the manuscript, rendering it suitable for acceptance in its present form.

Minor editing of English language required.

Author Response

Ponit 1: The authors have made substantial improvements to the manuscript, rendering it suitable for acceptance in its present form.

Response 1: Thank you for recognizing our research efforts.

Ponit 2: Minor editing of English language required.

Response 2: Kindly seek assistance from the editor to rectify it, or explicitly specify the required modifications.

This manuscript is a resubmission of an earlier submission. The following is a list of the peer review reports and author responses from that submission.

Round 1

Reviewer 1 Report

Firstly, it was very hard to read the manuscript, I started to fix some mistakes, but then I gave up as there is a problem nearly in each sentence; either the sentence is too short, the words used in the sentence are not the best choice or not used normally in that context, grammatically wrong or the author did not deliver the message (meaningless sentences), huge amount of typo errors, capital and Italics. Organisms name is wrongly written without following the scientific ruling. It is poorly written. Manuscript requires professional editing on a serious note. It looks like authors chose random synonyms to reduce the plagiarism without any coherence to the scientific meaning and context. I will give some examples from abstract alone. Unfortunately, I cannot spare enough time to write 5 pages of corrections.

1.      as alternative treatments or preventative therapies for a variety of bacterial infection diseases…..

preventive therapies for various infectious bacterial diseases

2.      Therefore, we reviewed presents some of the anti-virulence and inhibiting bacterial toxin strategies that are currently being developed

reviewed and presents cannot come in the same sentence. Moreover, what you mean by currently being developed. You have not provided any information related to development of vaccine or any agent targeting bacterial resistance.

3.      Ultimately, although further research is needed……

Ultimately, although further altogether in one line.

4.      before a definitive statement can be made….

It should be “before any conclusions can be made”

5.     a new tool in the arsenal in the fight

Arsenal????? Seriously???? Moreover, in the arsenal, in the fight together….

It’s no sense to incorporate two vastly described things into one. Probiotics and Medicinal plants are like oceans in science. Authors have merged meaninglessly in the review. What was author's motto or conceptual thinking to merge two completely different things into one without any relation and coherence. You should have stick to only one part either Probiotics or Medicinal plants and write more focused and in-depth review. There are thousands of literatures on importance of Medicinal plants targeting quorum sensing. Various bioactive components of plant extracts responsible of inhibiting various pathways in bacterial signaling. Authors have merely touched even 1% in this review.

Figure 1 is meaningless. Very basic. Even the spelling of reduction is not correctly written. Similarly, there is no mechanism in figure 4. A very well-known information.

In my view, the authors have severely failed in providing a convincing context and research gaps present in the current literature or problems related to the subject and why their current study is valuable and/or deserving of the reader's time and attention. All information is well-known and there is no novelty.

Writing is so confusing. There is not even 1 table. Authors should have provided two tables. One for list of probiotics, target, and mode of action. Similarly for plant secondary metabolites with list of plants extracted from and targets in bacterial quorum sensing. This would have made reading much better and attracts the reader’s attention.

At this stage manuscript is nowhere to be sent for revision. It must be rejected for thorough rewriting.

Author Response

Response to Reviewer 1 Comment

Ponit 1: Firstly, it was very hard to read the manuscript, I started to fix some mistakes, but then I gave up as there is a problem nearly in each sentence; either the sentence is too short, the words used in the sentence are not the best choice or not used normally in that context, grammatically wrong or the author did not deliver the message (meaningless sentences), huge amount of typo errors, capital and Italics. Organisms name is wrongly written without following the scientific ruling. It is poorly written. Manuscript requires professional editing on a serious note. It looks like authors chose random synonyms to reduce the plagiarism without any coherence to the scientific meaning and context. I will give some examples from abstract alone. Unfortunately, I cannot spare enough time to write 5 pages of corrections.

Response 1: Thank you for your careful review. We are very sorry for the mistakes in the manuscript and inconvenience they caused in your reading. The manuscript has been thoroughly revised and rewritten by MDPI English Editing. so we hope it can meet the journal’s standard.

Ponit 2: Figure 1 is meaningless. Very basic. Even the spelling of reduction is not correctly written.

Response 2: Thank you for your suggestion. We have revised the spelling.

Ponit 3:Authors should have provided two tables. One for list of probiotics, target, and mode of action. Similarly for plant secondary metabolites with list of plants extracted from and targets in bacterial quorum sensing. This would have made reading much better and attracts the reader’s attention.

Response 3:Thank you for your advice. We have added two tables.

Reviewer 2 Report

Dear Authors:

I checked the contents of your manuscript and confirmed that the potential paper was very informative and well-arranged. This will be an excellent reference for the readers. However, the conclusion part is not good.  

Please add the new section before the conclusions. The section title should be Discussion and describe your idea about the current research critically and future plan of this field. 

 You described it from line 492 to line 496, but it was hard to follow. For example, you wrote: "It is clear that medicinal plant resources and probiotics are abundant. Thus, many aspects have not been studied..."  However, the sentences don't sound logical. And the description is insufficient. And the sentences from lines 482 to 491 don't summarize your contents. And insufficient.  

Author Response

Response to Reviewer 2 Comment

Ponit 1: Please add the new section before the conclusions. The section title should be Discussion and describe your idea about the current research critically and future plan of this field. 

Response 1: Thank you for your careful review. We have added future plan of this field.

Ponit 2: You described it from line 492 to line 496, but it was hard to follow. For example, you wrote: "It is clear that medicinal plant resources and probiotics are abundant. Thus, many aspects have not been studied..."  However, the sentences don't sound logical. And the description is insufficient. And the sentences from lines 482 to 491 don't summarize your contents. And insufficient. 

Response 2: Thank you for your suggestion. We have revised these sentences.

Reviewer 3 Report

The review is interesting and well written. I have only small observations, after which the review is, in my opinion, ready for publication:

- Escherichia coli (line 41) should be written in italics

- MRSA (279) should be written in italics

- some parts of figure 5 are of low quality; they should be replaced;

- Author Contributions (line 496) is found by mistake in the conclusions...

it should be moved elsewhere.

Author Response

Response to Reviewer 3 Comment

Ponit 1:  Escherichia coli (line 41) should be written in italics.

Response 1: Thank you for your suggestion. We have revised.

Ponit 2:  MRSA (279) should be written in italics.

Response 2: Thank you for your suggestion. We have revised.

Ponit 3: some parts of figure 5 are of low quality; they should be replaced.

Response 3: Thank you for your suggestion. we have revised the spelling.

Ponit 4: Author Contributions (line 496) is found by mistake in the conclusions...

it should be moved elsewhere.

Response 4: Thank you for your suggestion. We have revised.